# Creation of chiral interface channels for quantized transport in magnetic topological insulator multilayer heterostructures

Yi-Fan Zhao [1,5], Ruoxi Zhang [1,5], Jiaqi Cai [2], Deyi Zhuo[1], Ling-Jie Zhou [1], Zi-Jie Yan [1], Moses H. W. Chan[1], Xiaodong Xu [2,3] & Cui-Zu Chang [1,4] ✉

One-dimensional chiral interface channels can be created at the boundary of two quantum anomalous Hall (QAH) insulators with different Chern numbers. Such a QAH junction may function as a chiral edge current distributer at zero magnetic field, but its realization remains challenging. Here, by employing an in-situ mechanical mask, we use molecular beam epitaxy to synthesize QAH insulator junctions, in which two QAH insulators with different Chern numbers are connected along a one-dimensional junction. For the junction between Chern numbers of 1 and −1, we observe quantized transport and demonstrate the appearance of the two parallel propagating chiral interface channels along the magnetic domain wall at zero magnetic field. For the junction between Chern numbers of 1 and 2, our quantized transport shows that a single chiral interface channel appears at the interface. Our work lays the foundation for the development of QAH insulator-based electronic and spintronic devices and topological chiral networks.

Topological materials are unique solid-state systems that exhibit topologically protected boundary states (i.e., edge/surface states). As a consequence of the intrinsic protection that prevents impurity scattering and allows for manipulations and measurements, these topological edge/surface states have been predicted to be useful for the next generation of quantum-based electronic and spintronic devices as well as topological quantum computations[1,2]. Over the past -15 years, topological band theory has played a key role in the discovery of new topological materials[1-4]. The interplay between the bulk topology and protected edge/surface states in topological materials is usually referred to as the bulk-boundary correspondence. In other words, the formation of the topological edge/surface states is guaranteed by the topological character of the bulk bands. In addition to the edge/surface states in naturally occurring topological materials, topologically protected interface states can also be engineered at the interfaces between two materials with different topological invariants.

The quantum anomalous Hall (QAH) insulator is a prime example of two-dimensional (2D) topological states and possesses dissipation-free chiral edge states (CESs) on its boundaries[4-10]. In QAH insulators, the Hall resistance is quantized at $h/e^2$ and the longitudinal resistance vanishes under zero magnetic field. The QAH effect was first realized in magnetically doped topological insulators (TI), specifically, Cr-doped and/or V-doped $(Bi, Sb)_2Te_3$ thin films[7,8,10-15]. More recently, the QAH effect was also observed in thin flakes of intrinsic magnetic TI $MnBi_2Te_4$ (Ref. [16]) and moiré materials formed from graphene[17] or transition metal dichalcogenides[18]. According to topological band theory, chiral interface channels (CICs) also appear at the interfaces between two QAH insulators with different Chern numbers $C$. The CIC number is determined by the difference in $C$ between these two adjacent QAH insulator domains. The CIC propagating direction (i.e., chirality) is dictated by the relative orientation of the spontaneous magnetization in the two QAH insulators[1,2,19]. Therefore, the creation and manipulation of CESs and/or CICs can facilitate the development of topological chiral networks[20,21], which have the potential for applications in energy-efficient QAH-based electronic and spintronic devices. Moreover, it has been proposed that chiral Majorana physics in

[1]Department of Physics, The Pennsylvania State University, University Park, PA 16802, USA. [2]Department of Physics, University of Washington, Seattle, WA 98195, USA. [3]Department of Material Science and Engineering, University of Washington, Seattle, WA 98195, USA. [4]Materials Research Institute, The Pennsylvania State University, University Park, PA 16802, USA. [5]These authors contributed equally: Yi-Fan Zhao, Ruoxi Zhang. ✉e-mail: cxc955@psu.edu

QAH/superconductor heterostructures can be probed by placing a grounded superconductor island on the domain boundary between $C = +1$ and $C = -1$ QAH insulators in a Mach-Zehnder interferometer configuration[22,23]. Magnetic force microscope (MFM)[24] and the Meissner effect of a bulk superconductor cylinder[25] have been employed to create a magnetic domain wall (DW) in QAH insulators, which is unfeasible for device fabrication. Therefore, the synthesis of a designer magnetic DW in a QAH insulator (i.e., a junction between $C = +1$ and $C = -1$ QAH insulators) and the junction between two QAH insulators with arbitrary $C$ are highly desirable with exceptional promise for potential topological circuit applications.

In this work, we synthesize QAH insulator junctions in magnetic TI/TI multilayer heterostructures by employing an in-situ mechanical mask in our MBE chamber. Our electrical transport measurements show quantized transport in these QAH insulator junctions, which indicates the appearance of the CICs near the magnetic DW. For the junction between $C = +1$ and $C = -1$ QAH insulators, we find two parallel propagating CICs at the magnetic DW. For the junction between $C = 1$ and $C = 2$ QAH insulators, one CES tunnels through the QAH DW entirely, while the second CIC propagates along the QAH DW. The number of CICs is determined by the difference in $C$ between the two QAH insulators. We show these QAH insulator junctions with robust CICs are feasible for device fabrication and thus provide a platform for the development of QAH-based electronic and spintronic devices and topological quantum computations.

## Results

### MBE growth and electrical transport measurements

All QAH junction samples are grown on heat-treated -0.5 mm thick SrTiO₃(111) substrates in a commercial MBE chamber (Omicron Lab10) (Method; Supplementary Figs. 1 to 3). The Bi/Sb ratio in each layer is optimized to tune the chemical potential of the sample near the charge neutral point[7,8,26–28]. The electrical transport measurements are carried out in a Physical Property Measurements System (Quantum Design DynaCool, 1.7 K, 9 T) and a dilution refrigerator (Leiden Cryogenics,

10 mK, 9 T) with the magnetic field applied perpendicular to the sample plane. The mechanically scratched Hall bars are used for electrical transport measurements. More details about the MBE growth and electrical transport measurements can be found in Methods.

### The junction between $C = +1$ and $C = -1$ QAH insulators

We first focus on the junction between $C = +1$ and $C = -1$ QAH insulators (Fig. 1a). To create this junction, we grow 2 quintuple layers (QL) $(Bi,Sb)_{1.74}Cr_{0.26}Te_3$/2 QL $(Bi,Sb)_2Te_3$/2 QL $(Bi,Sb)_{1.74}Cr_{0.26}Te_3$ sandwich heterostructure. Next, by placing an in-situ mechanical mask as close as possible to the sample surface (Supplementary Fig. 1), we deposit 2 QL $(Bi, Sb)_{1.78}V_{0.22}Te_3$ on one side of the sample. Since the coercive field $(\mu_0H_c)$ of V-doped $(Bi, Sb)_2Te_3$ films is much larger than that of Cr-doped $(Bi, Sb)_2Te_3$ films[8,29], the $\mu_0H_c$ of the Cr-doped $(Bi, Sb)_2Te_3$ sandwich layer is enhanced as a result of the existence of the interlayer exchange coupling[30–32]. We note that the middle 2 QL undoped $(Bi, Sb)_2Te_3$ layer is chosen here to couple the magnetizations of the two Cr-doped $(Bi, Sb)_2Te_3$ layers. As a consequence, the areas with (i.e., Domain II) and without (i.e., Domain I) 2 QL $(Bi, Sb)_{1.78}V_{0.22}Te_3$ possess different values of $\mu_0H_c$. When an external $\mu_0H$ is tuned between two $\mu_0H_c$s, an antiparallel magnetization alignment appears between Domain I (with $\mu_0H_{c1}$) and Domain II (with $\mu_0H_{c2}$). Therefore, a junction between $C = +1$ and $C = -1$ QAH insulators is created (Figs.1a, 1b, and Supplementary Figs. 4–10). Such a QAH DW can persist at zero magnetic field (Fig. 2d, e, and Supplementary Figs. 9, 10).

To characterize its magnetic property, we perform reflective magnetic circular dichroism (RMCD) measurements on the junction between $C = +1$ and $C = -1$ QAH insulators at $T = 2.5$ K (Fig. 1c–g, and Supplementary Fig. 4). The RMCD signals of both $C = +1$ and $C = -1$ QAH domains show hysteresis loops with different values of $\mu_0H_c$, confirming the ferromagnetic properties of the magnetic TI multilayers. The $\mu_0H_{c1}$ value of Domain I is -0.035 T, while the $\mu_0H_{c2}$ value of Domain II is -0.110 T (Fig. 1f, g). By mapping the samples under $\mu_0H$-0.075 T, between $\mu_0H_{c1}$ and $\mu_0H_{c2}$, we recognize Domain I and

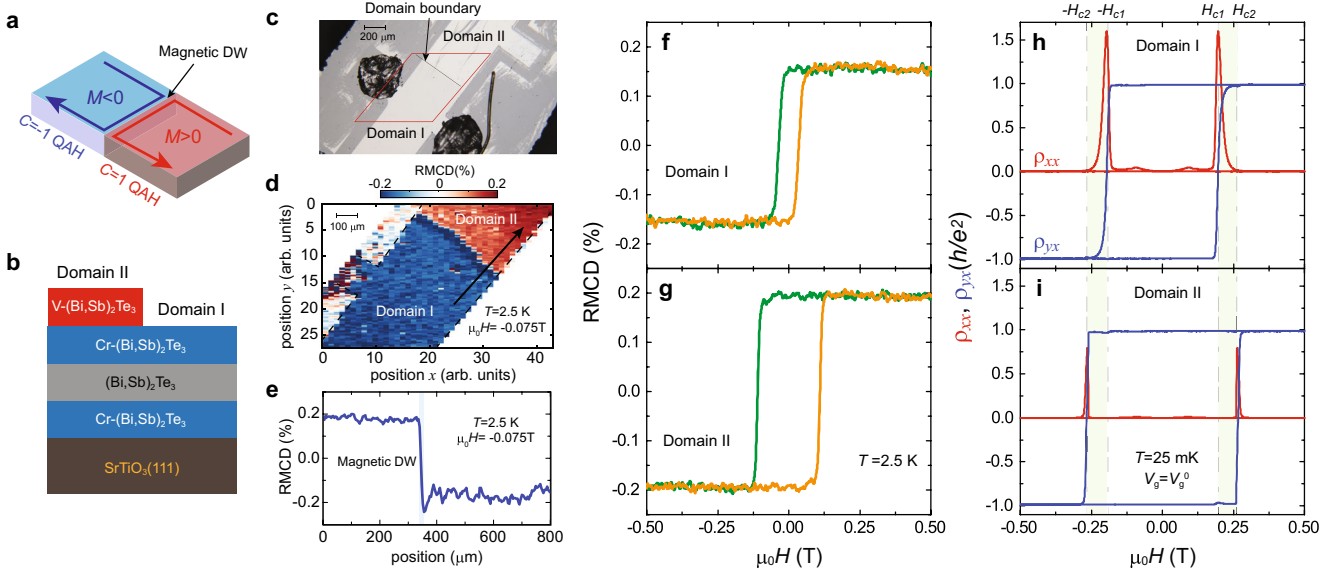

**Fig. 1 | Creation of magnetic domain walls (DWs) in quantum anomalous Hall (QAH) insulators. a** Schematic of chiral interface channels (CICs) near the magnetic DW in QAH insulators. $M > 0$ and $M < 0$ indicate upward (red) and downward (blue) magnetizations, respectively. **b** Side view of the magnetic TI multilayer structure for the junction between $C = 1$ QAH and $C = -1$ QAH insulators. The thickness of each layer is 2 QL. **c**, The optical image of the junction between $C = 1$ QAH and $C = -1$ QAH insulators. **d** Reflective magnetic circular dichroism (RMCD) map of the red rhomboid area in (**c**) measured at $\mu_0H = -0.075$ T and $T = 2.5$ K.

**e** RMCD signal measured along the arrow in (**d**) at $\mu_0H = -0.075$ T and $T = 2.5$ K. **f, g** $\mu_0H$ dependence of the RMCD signal of Domain I (**f**) and Domain II (**g**) measured at $T = 2.5$ K. **h, i** $\mu_0H$ dependence of $\rho_{xx}$ (red) and $\rho_{yx}$ (blue) of Domain I (**h**) and Domain II (**i**) measured at $V_g = V_g^0$ and $T = 25$ mK. Domain I: $\mu_0H_{c1}$ -0.195 T; Domain II: $\mu_0H_{c2}$ -0.265 T. The magnetic DW can be created by tuning $\mu_0H$ between $\mu_0H_{c1}$ and $\mu_0H_{c2}$. The data in (**h**) and (**i**) are symmetrized or anti-symmetrized as a function of $\mu_0H$ to eliminate the influence of the electrode misalignment.

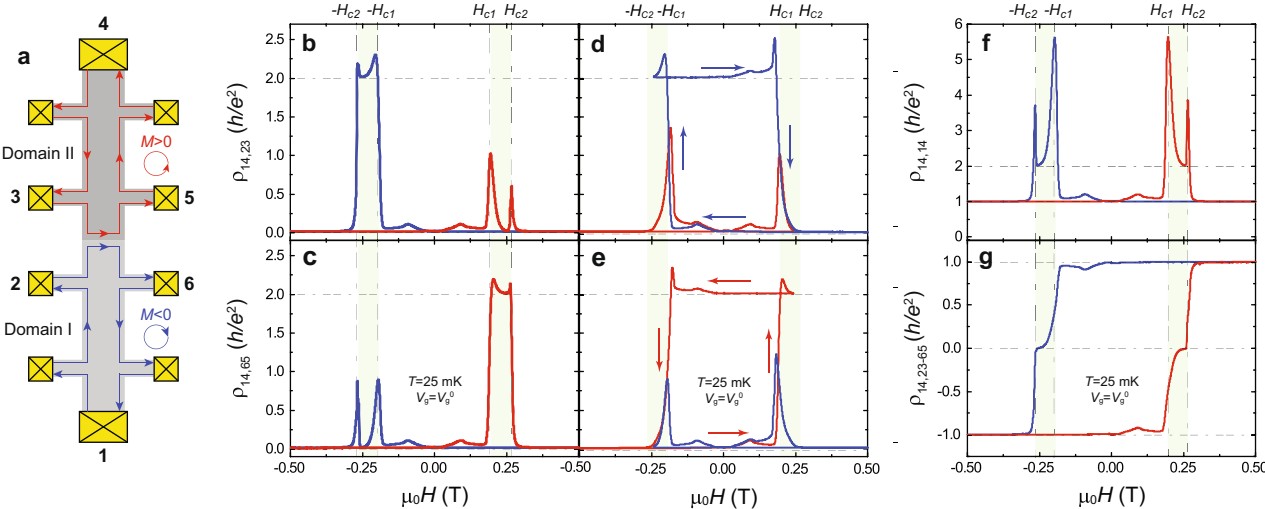

**Fig. 2 | Quantized transport along the junction between $C = 1$ QAH and $C = -1$ QAH insulators. a** Schematic of chiral edge/interface channels when the current flows from contact 1 to contact 4 (i.e., from Domain I to Domain II). The red and blue lines indicate the left- and right-handed chiral edge states with upward and downward magnetization, respectively. **b, c** $\mu_0 H$ dependence of $\rho_{14,23}$ (**b**) and $\rho_{14,65}$ (**c**). **d, e** Minor loops of $\rho_{14,23}$ (**d**) and $\rho_{14,65}$ (**e**). The arrows indicate the magnetic field sweep directions in minor loop measurements. For the minor loop shown in blue, the magnetic field is swept from + 0.50 T to −0.24 T, which is between $\mu_0 H_{c1} = -0.195$ T and $\mu_0 H_{c2} = -0.265$ T, and then swept back to + 0.50 T. For the minor loop shown in red, the magnetic field is swept from −0.50 T to +0.24 T, and then swept back to −0.50 T. **f** $\mu_0 H$ dependence of two-terminal resistance $\rho_{14,14}$. **g** $\mu_0 H$ dependence of the Hall resistance $\rho_{14,23-65}$. $\rho_{14,23-65}$ is measured between 2,3 and 6,5. Here contacts 2 and 3 are connected, same for contacts 5 and 6. All measurements are performed at $V_g = V_g^0$ and $T = 25$ mK.

Domain II. This further confirms that a magnetic DW is created at the boundary of Domain I and Domain II (Fig. 1d, e, and Supplementary Fig. 4). Next, we perform electrical transport measurements on Domain I and Domain II at $T = 25$ mK and at the charge neutral point $V_g = V_g^0$. Both domains show well quantized $C = 1$ QAH effect. For Domain I, the Hall resistance $\rho_{yx}$ under zero magnetic field [labeled as $\rho_{yx}(0)$] is ~0.988 $h/e^2$, concomitant with $\rho_{xx}(0)$ ~0.003 $h/e^2$ (~80 Ω) (Fig. 1h). For Domain II, $\rho_{yx}(0)$~0.985 $h/e^2$ and $\rho_{xx}(0)$ ~0.0008 $h/e^2$ (~20 Ω) (Fig. 1i). At $T = 25$ mK, the values of $\mu_0 H_{c1}$ and $\mu_0 H_{c2}$ are found to be ~0.195 T and ~0.265 T, respectively. Both values are much larger than those measured in RMCD at $T = 2.5$ K (Fig. 1f, g). Therefore, when $\mu_0 H_c$ is tuned between $\mu_0 H_{c1}$ and $\mu_0 H_{c2}$, Domain I and Domain II possess antiparallel magnetization alignment, and thus a junction between $C = +1$ and $C = -1$ QAH insulator is established.

Next, we perform magneto-transport measurements across the magnetic DW at $T = 25$ mK and $V_g = V_g^0$. The schematic of the Hall bar device is shown in Fig. 2a. The $\mu_0 H$ dependence of the longitudinal resistance across the magnetic DW $\rho_{14,23}$ and $\rho_{14,65}$ is shown in Fig. 2b, c, respectively. The red (blue) curves represent upward (downward) $\mu_0 H$ sweeps. When Domain I and Domain II have the parallel magnetization alignment, the entire sample behaves as a QAH insulator with 1D CESs along its edges and thus both $\rho_{14,23}$ and $\rho_{14,65}$ vanish. Since $\mu_0 H_c$ of Domain II (i.e., $\mu_0 H_{c2}$ ~0.265 T) is larger than that of Domain I (i.e., $\mu_0 H_{c1}$ ~0.195 T), sweeping external $\mu_0 H$ first reverses the magnetization of Domain I. Therefore, when $\mu_0 H$ is tuned between $\mu_0 H_{c1}$ and $\mu_0 H_{c2}$, the magnetizations of Domain I and Domain II are in antiparallel alignment and thus a magnetic domain boundary is formed in a $C = 1$ QAH insulator (Figs. 1a and 2a), consistent with our RMCD results on the same device (Fig. 1d, e, and Supplementary Fig. 4). For the Domain I-downward-Domain II-upward state, $\rho_{14,23}$~2.017 $h/e^2$ and $\rho_{14,65}$ ~0.024 $h/e^2$. However, for the Domain I-upward-Domain II-downward state, $\rho_{14,23}$ ~ 0.018 $h/e^2$ and $\rho_{14,65}$ ~ 2.009 $h/e^2$. During each magnetization reversal, the quantized transport via the dissipation-free CES in the corresponding domain fades away. Therefore, half of the sample becomes dissipative and a large longitudinal resistance peak appears. This is the reason for the two-peak feature in $\rho_{14,23}$ (Fig. 2b), $\rho_{14,65}$ (Fig. 2c), and $\rho_{14,14}$ (Fig. 2f). To examine the quantized transport across the magnetic DW in a $C = 1$ QAH insulator, we measure $\mu_0 H$

dependence of $\rho_{14,23}$ and $\rho_{14,65}$ at different gate voltages $V_g$ under different DW configurations (Supplementary Fig. 7). We note that these quantized transport behaviors can be well interpreted by the Landauer−Büttiker formalism based on the assumption that each CIC has ~50% transmission probability through the magnetic DW between Domain I and Domain II (Fig. 2a; Supplementary Note 4). Finally, we demonstrate that this artificial magnetic DW in a $C = 1$ QAH insulator persists at zero magnetic field by minor loop measurements (Fig. 2d, e, and Supplementary Figs. 9, 10).

Figure 2f shows the $\mu_0 H$ dependence of the two-terminal resistance $\rho_{14,14}$ at $T = 25$ mK and $V_g = V_g^0$. By tuning the magnetizations of Domain I and Domain II from parallel to antiparallel alignments, the value of $\rho_{14,14}$ is found to change from ~ $h/e^2$ to ~2 $h/e^2$, confirming the assumption that the transmission probability of the CIC through the magnetic DW is ~50%. Figure 2g shows the $\mu_0 H$ dependence of the Hall resistance $\rho_{14,23-65}$, which is measured by contact configurations 2,3 and 6,5. Here contacts 2 and 3 are connected, same for contacts 5 and 6. The observation of the zero Hall resistance plateau for $\mu_0 H_{c1} < \mu_0 H < \mu_0 H_{c2}$ validates the appearance of the two parallel CICs at the magnetic DW. The same behavior is also observed in the second device with the contacts directly sitting on the magnetic DW boundary (Supplementary Fig. 9f). We note that in this device the two-terminal resistance along the magnetic DW $\rho_{78,78}$ is ~ $h/2e^2$ under $\mu_0 H_{c1} < \mu_0 H < \mu_0 H_{c2}$, further validating the emergence of two parallel CICs at the magnetic DW between $C = +1$ and $C = -1$ QAH insulators (Supplementary Fig. 9g).

To demonstrate the current splitting function of the QAH junction, we perform current distribution measurements to directly detect the two parallel CICs at the junction interface. As shown by the red lines in Fig. 3a–d, we first inject a bias current $I$ of ~1 nA at contact 1 and measure the currents flowing to the ground through contacts 2 ($I_2$) and 3 ($I_3 = I - I_2$) with all other floating contacts. When Domain I and Domain II have negative parallel magnetization (i.e., $M < 0$) alignment, ~97% of the drain current is found to flow through contact 2, while ~3% of the drain current flows through contact 3 (Fig. 3a, e). The nonzero drain current through contact 3 is attributable to residual dissipation channels in the QAH insulator device[33,34]. However, when Domain I and Domain II have positive parallel magnetization (i.e., $M > 0$) alignment,

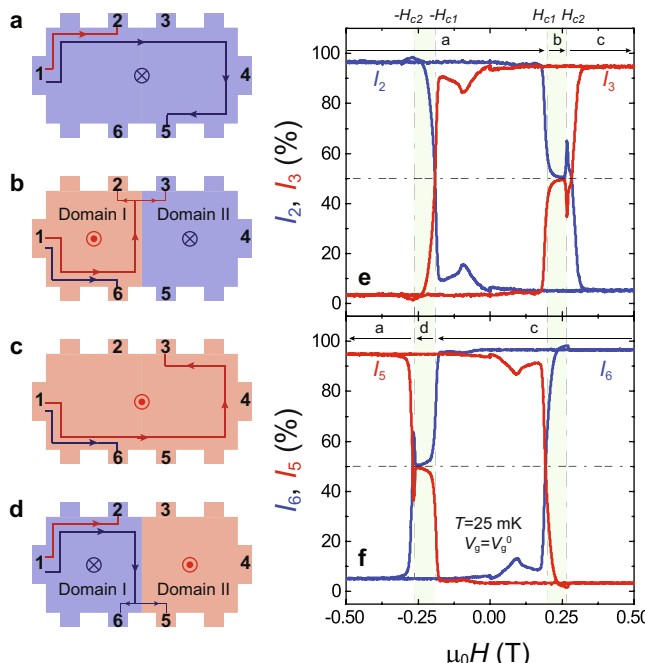

**Fig. 3 | Chiral edge current distributor of the junction between $C = 1$ QAH and $C = -1$ QAH insulators. a–d** Schematics of chiral edge/interface current under different DW configurations, which are created by tuning external $\mu_0 H$. When a current of -1 nA is injected from contact 1, the drain current measured at contact 2 or 3 with other floating contacts is shown in red, while the drain current measured at contact 5 or 6 with other floating contacts is shown in blue. **e** $\mu_0 H$ dependence of normalized drain current for contact 2 (blue) and 3 (red). **f** $\mu_0 H$ dependence of normalized drain current for contact 5 (red) and 6 (blue). All measurements are performed at $V_g = V_g^0$ and $T = 25$ mK. The arrows in (**e, f**) indicate the magnetic field sweep directions and label the DW configurations in (**a–d**).

~5% of the drain current flows through contact 2, while ~95% of the drain current flows through contact 3 (Fig. 3c, e). For $\mu_0 H_{c1} < \mu_0 H < \mu_0 H_{c2}$, i.e., under the Domain I-upward-Domain II-downward configuration, we find $I_2 = I_3$, which is direct evidence for the existence of two parallel propagating CICs at the magnetic DW (Fig. 3b, e). However, for $-\mu_0 H_{c2} < \mu_0 H < -\mu_0 H_{c1}$, i.e., under the Domain I-downward-Domain II-upward configuration, $I_2$ is no longer equal to $I_3$. Instead, $I_2$ and $I_3$ switch the dominance as soon as the magnetic DW appears (i.e., at $-\mu_0 H_{c1}$) (Fig. 3d, e). This is a result of the bias current directly flowing to contact 2 without passing through the magnetic DW under Domain I-downward-Domain II-upward configuration. Mirroring behaviors (Fig. 3f) are observed when we measure the currents flowing to the ground through contacts 5 ($I_5$) and 6 ($I_6 = I - I_5$) with all other floating contacts, as shown by the blue lines in Fig. 3a–d. The existence of the two parallel propagating CICs at the magnetic DW is further confirmed by injecting the bias current at contact 4 (Supplementary Figs. 8, 10).

### The junction between $C = 1$ and $C = 2$ QAH insulators

Since the Chern number $C$ of the QAH insulators in magnetic TI/TI multilayers can be tuned by altering the magnetic TI/TI bilayer periodicity[27,28], the junction between the two QAH insulators with two arbitrary $C$ can be achieved by growing different periods of magnetic TI/TI on the two sides of the sample. Here we use the junction between $C = 1$ and $C = 2$ QAH insulators as an example (Fig. 4 and Supplementary Figs. 11, 12). As shown in Fig. 4a, Domain I is 3QL $(Bi, Sb)_{1.74}Cr_{0.26}Te_3$/ 4QL $(Bi, Sb)_2Te_3$/3QL $(Bi, Sb)_{1.74}Cr_{0.26}Te_3$ sandwich heterostructure with the $C = 1$ QAH state, while Domain II is [3QL $(Bi, Sb)_{1.74}Cr_{0.26}Te_3$/ 4QL $(Bi, Sb)_2Te_3]_2$/3QL $(Bi, Sb)_{1.74}Cr_{0.26}Te_3$ penta-layer heterostructure

with the $C = 2$ QAH state[27,28]. We perform electrical transport measurements on this QAH junction with the Hall bar configuration as shown in Fig. 4b. We find that the Hall resistance $\rho_{14,26}$ of Domain I is -0.987 $h/e^2$ and $\rho_{14,35}$ of Domain II is -0.494 $h/e^2$ under zero magnetic field at $T = 25$ mK and $V_g = V_g^0$, confirming the $C = 1$ and $C = 2$ QAH states in Domain I and Domain II, respectively (Fig. 4c). After we establish the presence of this junction between $C = 1$ and $C = 2$ QAH insulators, we study its properties by measuring $\rho_{14,23}$ and $\rho_{14,65}$ across the QAH DW (Fig. 4b). For $M > 0$, $\rho_{14,23}$ is found to be -0.021 $h/e^2$ at zero magnetic field, concomitant with $\rho_{14,65}$ -0.526 $h/e^2$. However, for $M < 0$, $\rho_{14,23}$ -0.527 $h/e^2$ and $\rho_{14,65}$ -0.022 $h/e^2$ at zero magnetic field (Fig. 4d, e). In other words, the values of $\rho_{14,23}$ and $\rho_{14,65}$ always differ by $-h/2e^2$ for $M > 0$ or $+h/2e^2$ for $M < 0$ at $V_g = V_g^0$. This constant difference is further demonstrated in the $(V_g - V_g^0)$ dependence of $\rho_{14,23}(0)$ and $\rho_{14,65}(0)$ plots (Fig. 4f, g). In addition to the four-terminal resistance $\rho_{14,23}$ and $\rho_{14,65}$, we also measure the two-terminal resistance $\rho_{14,14}$ of the junction between $C = 1$ and $C = 2$ QAH insulators. The value of $\rho_{14,14}$ is found to be $-h/e^2$ in the well-defined magnetization regime (Supplementary Fig. 12). We note that the $\rho_{14,14}$ behavior of the junction between $C = 1$ and $C = 2$ QAH insulators is essentially similar to that of the individual $C = 1$ QAH insulator[33]. This observation confirms that only one chiral edge channel carries current from contact 1 to contact 4.

## Discussion

To further understand the property of the CIC at the DW between $C = 1$ and $C = 2$ QAH insulators, we assume that the transmission probability of the CIC through the DW between $C = 1$ and $C = 2$ QAH insulators is $P$. Based on the Landauer–Büttiker formalism[35], $\rho_{14,23} = \frac{1-P}{P}\frac{h}{e^2}$ and $\rho_{14,65} = \frac{2-P}{2P}\frac{h}{e^2}$ for $M > 0$ (Fig. 4b and Supplementary Fig. 13). We can see that the difference between $\rho_{65}$ and $\rho_{23}$ is always $-h/2e^2$, independent of the value of $P$. In our devices, the $C = 1$ and $C = 2$ QAH insulators have the same coercive field ($\mu_0 H_c$) and always have parallel magnetization alignment, so the two QAH insulators share the same CES chirality and thus $P$-1 (Fig. 4b). Therefore, the chiral edge current can flow through the DW between the $C = 1$ and $C = 2$ QAH insulators and $\rho_{14,23}(0)$-0 and $\rho_{14,65}(0)$-$h/2e^2$ for $M > 0$ (Fig. 4f).

We further perform current distribution measurements on the junction between the $C = 1$ and $C = 2$ QAH insulators. By injecting a bias current $I$ of ~1 nA at contact 1 or 4, we measure the currents flowing to the ground through contacts 5 ($I_5$) and 6 ($I_6 = I - I_5$) with all other floating contacts. When the bias current is injected through contact 1, only one CES carries current from contact 1 to contact 4 (Fig.4b and Supplementary Fig. 12), so $I_5$ and $I_6$ switch the dominance during reversing $M$ (Fig. 4h). However, when the bias current is injected through contact 4, two CESs carry the current. For $M > 0$, one CES passes through the $C = 1$ QAH domain entirely and flows through contact 6, while the other CES travels along the DW and flows through contact 5. Therefore, the value of $I_5$ is equal to $I_6$ for $M > 0$, supporting the transmission probability of the CIC through the DW between $C = 1$ and $C = 2$ QAH insulators $P$-1. For $M < 0$, both CESs of the $C = 2$ QAH insulator carry current from contact 4 to contact 5 clockwise, and thus the value of $I_5$ is much larger than $I_6$ (Fig. 4i). Therefore, the junction between $C = 1$ and $C = 2$ QAH insulators acts as a chiral edge current divider at zero magnetic field.

To summarize, by placing an in-situ mask inside the MBE chamber, we show it is possible to grow two QAH insulators with different Chern numbers separated by a well-defined 1D junction in the form of a DW. Through systematic magneto-resistance measurements with different contact configurations that show the expected quantized and dissipation-free transport with appropriate current distribution, we demonstrate the creation of CICs at the DW between two QAH insulators and find that the number of CICs is determined by the Chern number difference between the two QAH insulators. For the junction between $C = +1$ and $C = -1$ QAH insulators, two parallel CICs propagate

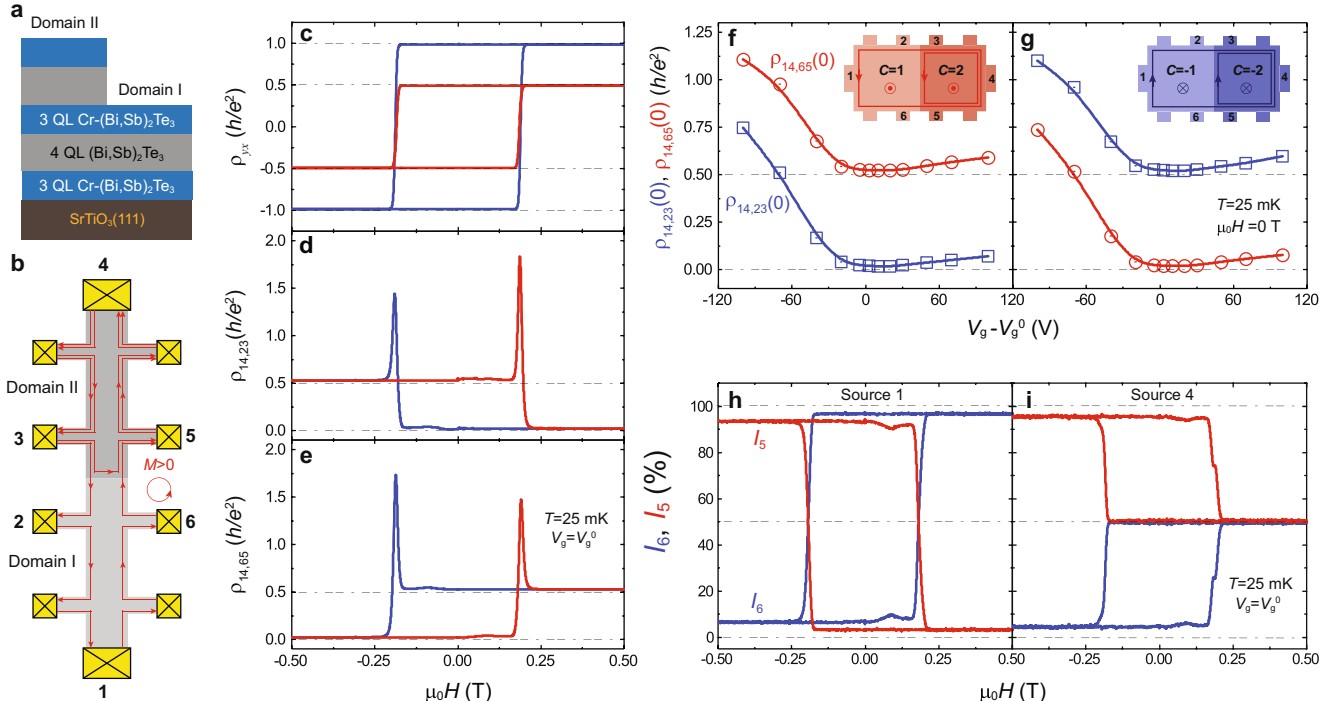

**Fig. 4 | Creation of CICs along the junction between $C = 1$ QAH and $C = 2$ QAH insulators. a** Side view of the magnetic TI multilayer structures for the junction between $C = 1$ QAH and $C = 2$ QAH insulators. **b** Schematic of the chiral edge/interface channels of the junction between $C = 1$ QAH and $C = 2$ QAH insulators for $M > 0$. **c** $\mu_0 H$ dependence of $\rho_{yx}$ for Domain I ($C = 1$, blue) and Domain II ($C = 2$, red). **d**, **e** $\mu_0 H$ dependence of $\rho_{14,23}$ (**d**) and $\rho_{14,65}$ (**e**). **f**, **g** Gate ($V_g$-$V_g^0$) dependence of $\rho_{14,23}(0)$ (blue) and $\rho_{14,65}(0)$ (red) for $M > 0$ (**f**) and $M < 0$ (**g**). **h**, **i** $\mu_0 H$ dependence of normalized drain current for contact 5 (red) and 6 (blue) with a bias current injected from contact 1 (**h**) or contact 4 (**I**). All measurements are performed at $V_g = V_g^0$ and $T = 25$ mK.

along the magnetic DW. For the junction between $C = 1$ and $C = 2$ QAH insulators, one CES tunnels through the QAH DW entirely and the other CIC propagates along the QAH DW. Our work provides a comprehensive understanding of the CES/CIC behavior in QAH insulators and advances our knowledge of the interplay between two QAH insulators. Moreover, the synthesis of the junction between two QAH insulators with different Chern numbers provides a unique opportunity to develop a transformative information technology based on the dissipation-free CES/CIC in QAH insulators.

## Methods

### MBE growth

The QAH insulator junctions (i.e., where two QAH insulators with different $C$ are connected) used in this work are fabricated by using an in-situ mechanical mask in a commercial MBE system (Omicron Lab10) with a base vacuum better than ~$2 \times 10^{-10}$ mbar. The sharp boundary between two QAH insulators is achieved by placing the in-situ mechanical mask as close as possible to the sample surface. All magnetic TI multilayer heterostructures are grown on heat-treated ~0.5 mm thick SrTiO₃(111) substrates. Before the growth of the QAH insulator junctions, the heat-treated SrTiO₃(111) substrates are first outgassed at ~600 °C for 1 h. Next, high-purity Bi (99.9999%), Sb (99.9999%), Cr (99.999%), V (99.999%), and Te (99.9999%) are evaporated from Knudsen effusion cells. During the growth of the magnetic TI multilayers, the substrate is maintained at ~230 °C. The flux ratio of Te per (Bi + Sb + Cr/V) is set to be greater than 10 to prevent Te deficiency in the films. The Bi/Sb ratio in each layer is optimized to tune the chemical potential of the entire magnetic TI multilayer heterostructure near the charge neutral point. The growth rate of both magnetic TI and TI films is ~0.2 QL per minute.

The in-situ mechanical mask is made from a 0.1 mm thick tantalum (Ta) foil with a physical dimension of ~2 mm × 10 mm. The Ta foil

mask is mounted on a custom-designed flag-style sample holder with a Ta screw (Supplementary Fig. 1a). Before the MBE growth, the Ta foil mask is placed in a position where the heat-treated SrTiO₃(111) is uncovered. A few periods of magnetic TI/TI multilayers are first grown to form Domain I. Next, a magnetic arm is used to rotate the Ta foil mask ~90 degrees to cover half of the sample and then additional periods of magnetic TI/TI multilayers are grown to form Domain II (Supplementary Fig. 1b).

### Electrical transport measurements

All QAH insulator junction samples for electrical transport measurements are grown on 2 mm × 10 mm insulating SrTiO₃(111) substrates are scratched into a Hall bar geometry with multiple pins (Figs. 2a, 4b) using a computer-controlled probe station. The width of the Hall bar is ~0.5 mm. The electrical ohmic contacts are made by pressing indium dots on the films. The bottom gate is prepared by flattening the indium dots on the back side of the SrTiO₃(111) substrates. Transport measurements are conducted using a Physical Property Measurement System (Quantum Design DynaCool, 1.7 K, 9 T) for $T \geq 1.7$ K and a Leiden Cryogenics dilution refrigerator (10 mK, 9 T) for $T < 1.7$ K. The excitation currents are 1 µA and 1 nA for the PPMS and the dilution measurements, respectively. Unless otherwise noted, the magneto-transport results shown in this work are the raw data. More transport results are found in Supplementary Figs. 5 to 12.

### RMCD measurements

The RMCD measurements on MBE-grown QAH insulator junction samples are performed in a closed-cycle helium cryostat (Quantum Design Opticool) at $T$ ~2.5 K and an out-of-plane magnetic field up to 0.5 T. A ~633 nm laser is used to probe the samples at normal incidence with the fixed power of ~1 µW. The AC lock-in measurement technique is used to measure the RMCD signals. The RMCD map is taken by stepping an attocube nanopositioner. The experimental setup

has been used in our prior measurements on MBE-grown $MnBi_2Te_4$ films[36].

## Data availability

The datasets generated during and/or analyzed during this study are available from the corresponding author upon request.

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

## Acknowledgements

We thank Yongtao Cui and Weida Wu for helpful discussions. This work is primarily supported by the AFOSR grant (FA9550-21-1-0177), including MBE growth, device fabrication, and RMCD measurements. The PPMS measurements are partially supported by the ARO Award (W911NF2210159). The dilution measurements and the data analysis are partially supported by the NSF-CAREER award (DMR-1847811). C.Z.C. also acknowledges the support from the Gordon and Betty Moore Foundation's EPiQS Initiative (Grant GBMF9063 to C.Z.C.).

## Author contributions

C.-Z.C. conceived and supervised the experiment. Y.-F.Z., D.-Y.Z., and Z.-J.Y. grew all QAH junction samples and carried out the PPMS transport measurements. R.Z., D.-Y.Z., L.-J.Z., and M.H.W.C. performed the dilution measurements. J.C. and X.X. performed the RMCD measurements. Y.-F.Z. and C.-Z.C. analyzed the data and wrote the manuscript with input from all authors.

## Competing interests

The authors declare no competing interests.
