## [Peer Review File · Nature Communications]

Creation of Chiral Interface Channels for Quantized Transport in Magnetic Topological Insulator Multilayer HeterostructuresREVIEWER COMMENTS

Reviewer #1 (Remarks to the Author):

Zhao et al create the magnetic domain wall (DW) in quantum anomalous Hall (QAH) insulators grown by molecular beam epitaxy with in-situ mechanical mask. The chiral interface channels (CICs) is realized at the DW, and the number of CICs is determined by the Chern number (C) difference between the two QAH insulators. The two parallel CICs propagate along the DW for the junction between $C = +1$ and $C = -1$ QAH insulators, while a single CIC appears at the interface for the junction between $C = 1$ to $C = 2$ QAH insulators. The quantized transport is well interpreted by the Landauer–Büttiker formalism. This work deepens the understanding of CIC behavior in QAH insulators. And the manuscript is clearly written. I recommend its publication in Nature Communications after addressing the following issues.

1) The two-terminal resistance $\rho_{14,14}$ was measured in the junction between $C = +1$ and $C = -1$ QAH insulators. How does the two-terminal resistance $\rho_{14,14}$ behave in the junction between $C = 1$ to $C = 2$ QAH insulators?

2) In a previous work (Ref. 28), the DW in QAH insulator is created by the magnetic force microscope, and the thickness of the sample is uniform. In this work, there is a sharp boundary between two QAH insulators. Would the nonuniform thickness influence the chiral interface channels in the junction?

3) The quantized transport of the QAH insulator usually appears at the charge neutral point, such as the data in Figs. 4f and 4g. However, the quantized transport seems to exist in a large range of gate voltages (about 100 V) in Supplementary Fig. 6. This behavior should be discussed in detail.

4) The RMCD maps at different magnetic field are shown in Supplementary Fig. 3. Please illustrate the scanning direction of the magnetic field.

Reviewer #2 (Remarks to the Author):

The authors demonstrated the ability to fabricate devices with designed domain walls between magnetic topological insulator films with different coercive fields or different topological band structures using MBE and an in-situ mechanical mask. They performed low temperature transport measurements to confirm the quantum anomalous Hall state of each side of the domain and the transport across the domain boundary, using Landauer-Buttiker formalism to understand the properties of the edge and boundary states. To my mind, these results are well supported by their data and analysis. I do not see any flaws in their data interpretation or conclusions.

I believe the work is novel and demonstrates remarkable control over the topological band structure and magnetism in these materials. The ability to reliably fabricate domain walls between different Chern insulators and study their properties is a meaningful step forward in studying the physics of the boundary states as well as towards useful topological circuits. I found the paper overall to be very well written and succinct, however there are a couple points where I'd like to see more information given. First, the use of a mechanical mask is integral to what is new, about this work, however very little information about what type of mask, how it is positioned and integrated into the MBE system, etc. In the interest of the ability to reproduce the experiment, more information about the mask should be provided.

Second, in figures where the field is swept in minor loops (Fig. 2 d,e) it is difficult to sort out how the field is being swept. Especially for readers who are not experts, it would be useful to describe in more detail how the fields are being swept for these measurements, with indicators on the figures, such as arrows showing the direction of field sweep.

With these minor additions, it is my believe that this work is well suited for publication in Nature Communications.

Reviewer #3 (Remarks to the Author):

The manuscript reported creation of chiral interface channels at the junction between magnetic topological insulator domains with different Chern numbers. The authors fabricated two junctions, including one junction between $C = 1$ and $C = -1$ QAH insulators and one junction between $C = 1$ and $C = 2$ QAH insulators. They found the number of propagating chiral interface channels are exactly determined by the bulk-boundary correspondence. Even though the topological current divider has been facilitated by MnBi₂Te₄ junction in a recent work "Dmitry Ovchinnikov et al., Nature Communications 13, 5967 (2022)", I find the strategy exploited in this manuscript shows more technological advances compared to the above reference. First, the sample was grown by MBE in this work, which means better reproducibility and larger sample size than junction made by MnBi₂Te₄ flakes (samples in this work are millimeter size while samples in MnBi₂Te₄ work are tens of micron meter size). More importantly, the in-situ mask technique used in this work is much more controllable compared to natural crystal edges in MnBi₂Te₄ flakes, which means it opens the possibility to fabricate artificial devices based on the topological current divider. Considering the technological novelties mentioned above, I recommend the publication of this manuscript in Nature Communications.

However, I found there is room to improve the presentations in this manuscript. I have several suggestions for the authors to consider:

1, In Figure 2, it is hard to figure out the sweeping direction of the magnetic field in a straightforward way, especially for the minor loops. It is preferable to label the sweeping direction on these resistivity curves.

2, The authors should briefly explain the appearance of two peaks in Figure 2 b and c, maybe also in Figure 2f. The underlying mechanism is not so clear to me.

3, In Figure 3, it would be better if the authors can label the corresponding stages (a, b, c, d) on the curves in Figure 3 e and f. This can help to understand how the transition from one stage to another happens.

-----Response to reviewers' comments-----

Reviewer #1 (Remarks to the Author):

Zhao et al create the magnetic domain wall (DW) in quantum anomalous Hall (QAH) insulators grown by molecular beam epitaxy with in-situ mechanical mask. The chiral interface channels (CICs) is realized at the DW, and the number of CICs is determined by the Chern number (C) difference between the two QAH insulators. The two parallel CICs propagate along the DW for the junction between $C = +1$ and $C = -1$ QAH insulators, while a single CIC appears at the interface for the junction between $C = 1$ to $C = 2$ QAH insulators. The quantized transport is well interpreted by the Landauer–Büttiker formalism. This work deepens the understanding of CIC behavior in QAH insulators. And the manuscript is clearly written. I recommend its publication in Nature Communications after addressing the following issues.

We thank Reviewer #1 for his/her positive assessment of our work and his/her recommendation for publication in Nature communications.

Comment 1:

1) The two-terminal resistance $\rho_{14,14}$ was measured in the junction between $C = +1$ and $C = -1$ QAH insulators. How does the two-terminal resistance $\rho_{14,14}$ behave in the junction between $C = 1$ to $C = 2$ QAH insulators?

Response: We did measure the two-terminal resistance $\rho_{14,14}$ of the junction between $C=1$ and $C=2$ QAH insulators at $T = 25$ mK and $V_g = V_g^0$ (Fig. R1). The value of $\rho_{14,14}$ is found to be $\sim h/e^2$ in the well-defined magnetization regime (Fig. R1c), which is consistent with our calculated value based on the Landauer–Büttiker formalism (Section II.4 of Supplementary Information). We noted that the $\rho_{14,14}$ behavior of the junction between $C=1$ and $C=2$ QAH insulators is essentially similar to that of the individual $C=1$ QAH insulator [Chang et al, Phys. Rev. Lett. 115, 057206 (2015)]. This observation further confirms that only one chiral edge channel carries current from contact 1 to contact 4 (Figs. R1a and R1b). We added Fig. R1 in the revised Supplementary Information and relevant discussion in the revised manuscript.

Fig. R1| Two-terminal resistance $\rho_{14,14}$ of the junction between $C = 1$ QAH and $C = 2$ QAH insulators. **a, b**, Schematic of the chiral edge/interface channels of the junction between $C = 1$ QAH and $C = 2$ QAH insulators for $M > 0$ (a) and $M < 0$ (b), respectively. **c**, $\mu_0 H$ dependence of $\rho_{14,14}$ measured at $T = 25$ mK and $V_g = V_g^0$. The charge neutral point $V_g^0 \sim +3$ V. (a) is reused here from Fig. 4b of the main text.

Comment 2:

2) In a previous work (Ref. 28), the DW in QAH insulator is created by the magnetic force microscope, and the thickness of the sample is uniform. In this work, there is a sharp boundary between two QAH insulators. Would the nonuniform thickness influence the chiral interface channels in the junction?

Response: We thank Reviewer #1 for bringing up this issue. For the junction between $C = +1$ and $C = -1$ QAH insulators (Fig. 1b of the main text), since both QAH insulators have one chiral edge state (CES), the additional 2 QL $(\text{Bi, Sb})_{1.78}\text{V}_{0.22}\text{Te}_3$ layer in Domain II is unlikely to influence the two chiral interface channels (CICs) near the magnetic domain wall (DW) (Figs. 1a and 2a of the main text). The observation of the $\sim 50\%$ transmission probability of the CIC confirms that the two CICs near the magnetic DW are equivalent (Fig. 2f of the main text). For the junction between $C = 1$ and $C = 2$ QAH insulators, the additional 3 QL $(\text{Bi, Sb})_{1.74}\text{Cr}_{0.26}\text{Te}_3/4$ QL $(\text{Bi, Sb})_2\text{Te}_3$ layer in Domain II (i.e., the $C = 2$ QAH insulator) creates the second CES, which propagates along the DW between $C = 1$ and $C = 2$ QAH insulators. Therefore, the nonuniform thickness of the QAH junction is also unlikely to influence the single CIC near the DW between $C = 1$ and $C = 2$ QAH

insulators (Fig.4b of the main text). We added this discussion in the revised Supplementary Information.

Comment 3:

3. The quantized transport of the QAH insulator usually appears at the charge neutral point, such as the data in Figs. 4f and 4g. However, the quantized transport seems to exist in a large range of gate voltages (about 100 V) in Supplementary Fig. 6. This behavior should be discussed in detail.

Response: We thank Reviewer #1 for raising this question. For magnetically doped TI thin films, the charge neutral point (i.e. $V_g=V_g^0$) is close to the bulk valence band maximum, but far away from the bulk conduction band minimum [see Fig. 5o of Chang et al, Phys. Rev. Lett.115, 057206 (2015)]. This asymmetric band structure results in the different gate-dependent transport behaviors between $V_g>V_g^0$ and $V_g<V_g^0$ [Chang et al, arXiv:2202.13902 (2022); Chang et al, Phys. Rev. Lett.115, 057206 (2015); Li et al, Sci. Rep. 6, 32732 (2016); Wang et al, Nat. Phys. 14, 791 (2018)]. For $V_g<V_g^0$, the chemical potential first crosses the bulk valence bands, and thus plenty of carriers are introduced, which leads to a large deviation from the QAH state. However, for $V_g>V_g^0$, the chemical potential first crosses the helical surface states, which introduces far fewer carriers and thus does not affect the QAH state as much. This is the reason that the quantized transport persists in a large range of gate voltage for $V_g>V_g^0$ [Chang et al, arXiv:2202.13902 (2022); Chang et al, Phys. Rev. Lett.115, 057206 (2015)]. We added this discussion in the revised Supplementary Information.

Comment 4:

4. The RMCD maps at different magnetic field are shown in Supplementary Fig. 3. Please illustrate the scanning direction of the magnetic field.

Response: We apologize to Reviewer #1 for the confusion. Our RMCD maps were obtained under the following scanning direction of the magnetic field: +0.500 T to +0.100 T (Supplementary Fig.4a), +0.100 T to -0.075 T (Supplementary Fig.4b), -0.075 T to -0.200 T (Supplementary Fig.4c), and -0.200T to +0.075T (Supplementary Fig.4d). We added this information in the caption of Supplementary Fig. 4.

Reviewer #2 (Remarks to the Author):

The authors demonstrated the ability to fabricate devices with designed domain walls between magnetic topological insulator films with different coercive fields or different topological band structures using MBE and an in-situ mechanical mask. They performed low temperature transport measurements to confirm the quantum anomalous Hall state of each side of the domain and the transport across the domain boundary, using Landauer-Buttiker formalism to understand the properties of the edge and boundary states. To my mind, these results are well supported by their data and analysis. I do not see any flaws in their data interpretation or conclusions.

I believe the work is novel and demonstrates remarkable control over the topological band structure and magnetism in these materials. The ability to reliably fabricate domain walls between different Chern insulators and study their properties is a meaningful step forward in studying the physics of the boundary states as well as towards useful topological circuits.

I found the paper overall to be very well written and succinct, however there are a couple points *where I'd like to see* more information given.

We thank Reviewer #2 for his/her concise summary and positive assessment of our work.

Comment 1:

First, the use of a mechanical mask is integral to what is new, about this work, however very little information about what type of mask, how it is positioned and integrated into the MBE system, etc. In the interest of the ability to reproduce the experiment, more information about the mask should be provided.

Response: We thank Reviewer #2 for bringing up this issue. The in-situ mechanical mask is made from a 0.1mm thick tantalum (Ta) foil with a physical dimension of $\sim 2\text{mm} \times 10\text{mm}$. The Ta foil mask is mounted on a custom-designed flag-style sample holder with a Ta screw (Fig. R2a). Before the MBE growth, the Ta foil mask is placed in a position where the heat-treated $\text{SrTiO}_3(111)$ is uncovered. We first grew a few periods of magnetic TI/TI multilayers to form Domain I. Next, we used a magnetic arm to rotate the Ta foil mask ~ 90 degrees to cover half of the sample and then grew additional periods of magnetic TI/TI multilayers to form Domain II (Fig. R2b). We added Fig. R2 in the revised Supplementary Information and relevant discussion in the Methods section of the revised manuscript.

Fig. R2| The *in-situ* mechanical mask used in our experiments. a, A custom-designed flag-style sample holder with a Ta foil mask. A 2 mm ×10 mm heat-treat SrTiO₃(111) is mounted in this sample holder. **b,** The QAH insulator junction is synthesized in magnetic TI/TI multilayer heterostructures by employing the *in-situ* Ta foil mask to cover half of the sample.

Comment 2:

Second, in figures where the field is swept in minor loops (Fig. 2 d,e) it is difficult to sort out how the field is being swept. Especially for readers who are not experts, it would be useful to describe in more detail how the fields are being swept for these measurements, with indicators on the figures, such as arrows showing the direction of field sweep.

Response: We thank Reviewer #2 for his/her suggestion. For the minor loop shown in blue, the magnetic field is swept from +0.50 T to -0.24 T, which is between $\mu_0 H_{c1} = -0.195\text{T}$ and $\mu_0 H_{c2} = 0.265\text{T}$, and then swept back to +0.50 T. For the minor loop shown in red, the magnetic field is swept from -0.50 T to +0.24 T, and then swept back to -0.50 T. We added arrows in Figs. 2d and 2e and Supplementary Figs. 9f, 10e, and 10f to show the direction of the magnetic field sweep with explanations in the figure captions.

With these minor additions, it is my believe that this work is well suited for publication in Nature Communications.

We thank Reviewer #2 for his/her recommendation for publication in *Nature communications*.

Reviewer #3 (Remarks to the Author):

The manuscript reported creation of chiral interface channels at the junction between magnetic topological insulator domains with different Chern numbers. The authors fabricated two junctions, including one junction between $C = 1$ and $C = -1$ QAH insulators and one junction between $C = 1$ and $C = 2$ QAH insulators. They found the number of propagating chiral interface channels are exactly determined by the bulk-boundary correspondence. Even though the topological current divider has been facilitated by MnBi₂Te₄ junction in a recent work “Dmitry Ovchinnikov et al., *Nature Communications* 13, 5967 (2022)”, I find the strategy exploited in this manuscript shows more technological advances compared to the above reference. First, the sample was grown by MBE in this work, which means better reproducibility and larger sample size than junction made by MnBi₂Te₄ flakes (samples in this work are millimeter size while samples in MnBi₂Te₄ work are tens of micron meter size). More importantly, the in-situ mask technique used in this work is much more controllable compared to natural crystal edges in MnBi₂Te₄ flakes, which means it opens the possibility to fabricate artificial devices based on the topological current divider. Considering the technological novelties mentioned above, I recommend the publication of this manuscript in *Nature Communications*.

However, I found there is room to improve the presentations in this manuscript. I have several suggestions for the authors to consider:

We thank Reviewer #3 for his/her positive assessment of our work and thoughtful comments.

Comment 1:

1. In Figure 2, it is hard to figure out the sweeping direction of the magnetic field in a straightforward way, especially for the minor loops. It is preferable to label the sweeping direction on these resistivity curves.

Response: We thank Reviewer #3 for his/her suggestion. We added arrows in Figs. 2d and 2e and Supplementary Figs. 9f, 10e, and 10f to show the direction of the magnetic field sweep. Please also see our response to Comment 2 of Reviewer #2 above.

Comment 2:

2. The authors should briefly explain the appearance of two peaks in Figure 2 b and c, maybe also in Figure 2f. The underlying mechanism is not so clear to me.

Response: We apologize to Reviewer #3 for the confusion. As noted in our manuscript, since Domain I and Domain II have different values of coercive field (μ_0H_c), these two domains reverse their magnetization at different μ_0H_c s when an external μ_0H is swept. During each magnetization reversion, the quantized transport via the dissipation-free CES in the corresponding domain fades away. Therefore, half of the sample becomes dissipative and a large longitudinal resistance peak appears. This is the primary reason for the two-peak feature in $\rho_{14,23}$ (Fig. 2b), $\rho_{14,65}$ (Fig. 2c), and $\rho_{14,14}$ (Fig. 2f). We added this discussion in the revised manuscript.

Comment 3:

3. In Figure 3, it would be better if the authors can label the corresponding stages (a, b, c, d) on the curves in Figure 3 e and f. This can help to understand how the transition from one stage to another happens.

Response: Done. In addition to Figs. 3e and 3f, we also labeled the other chiral edge current distribution curves in Supplementary Figs. 8e, 8f, and 10e to 10h.

-----List of changes-----

(All the changes in the main article are shown in blue)

1. Line 186 on Page 9, we added the below sentences.

“In addition to the four-terminal resistance $\rho_{14,23}$ and $\rho_{14,65}$, we also measure the two-terminal resistance $\rho_{14,14}$ of the junction between C=1 and C=2 QAH insulators. The value of $\rho_{14,14}$ is found to be $\sim h/e^2$ in the well-defined magnetization regime (Supplementary Fig. 12). We note that the $\rho_{14,14}$ behavior of the junction between C=1 and C=2 QAH insulators is essentially similar to that of the individual C=1 QAH insulator³⁵. This observation confirms that only one chiral edge channel carries current from contact 1 to contact 4.”

2. Line 244 on Page 11, we added the below paragraph.

“The in-situ mechanical mask is made from a 0.1mm thick tantalum (Ta) foil with a physical dimension of $\sim 2\text{mm} \times 10\text{mm}$. The Ta foil mask is mounted on a custom-designed flag-style sample holder with a Ta screw (Supplementary Fig. 1a). Before the MBE growth, the Ta foil mask is placed in a position where the heat-treated SrTiO₃(111) is uncovered. A few periods of magnetic TI/TI multilayers are first grown to form Domain I. Next, a magnetic arm is used to rotate the Ta foil mask ~ 90 degrees to cover half of the sample and then additional periods of magnetic TI/TI multilayers are grown to form Domain II (Supplementary Fig. 1b).”

3. Line 318 on Page 15, we added the below sentences to the caption of Fig. 2.

“The arrows indicate the magnetic field sweep directions in minor loop measurements. For the minor loop shown in blue, the magnetic field is swept from +0.50 T to -0.24 T, which is between $\mu_0 H_{c1} = -0.195\text{T}$ and $\mu_0 H_{c2} = -0.265\text{T}$, and then swept back to +0.50 T. For the minor loop shown in red, the magnetic field is swept from -0.50 T to +0.24 T, and then swept back to -0.50 T.”

4. Line 340 on Page 16, we added the below sentences to the caption of Fig. 3.

“The arrows in (e and f) indicate the magnetic field sweep directions and label the DW configurations in (a to d).”

5. We added Sections II.2 to II.3 in Supplementary Information.

6. We added Figs. S1 and S12 in Supplementary Information.

7. We added three references shown in blue in the revised Supplementary Information.

8. We made numbers of minor revisions shown in blue in the revised manuscript and Supplementary Information.

REVIEWERS' COMMENTS

Reviewer #1 (Remarks to the Author):

The authors have properly addressed the reviewers' comments and I recommend its publication in Nature Communications.

Reviewer #3 (Remarks to the Author):

In the revised manuscript, the authors have properly addressed all my previous concerns about some presentation issues in the original manuscript.

Considering the efforts that the authors have devoted to this revised manuscript and the significantly improved quality of the manuscript, I recommend this work be published in Nature Communications.